# Antigenotoxic and Antimutagenic Potentials of Proline in *Allium cepa* Exposed to the Toxicity of Cadmium

**Cornelia Purcarea** [1],[†] , **Vasile Laslo** [2],[*],[†] , **Adriana Ramona Memete** [3],[†] , **Eliza Agud** [2],[†] , **Florina Miere (Groza)** [4],[†] and **Simona Ioana Vicas** [1],[*],[†]

1   Department of Food Engineering, Faculty of Environmental Protection, University of Oradea, 26 Gen. Magheru Street, 410048 Oradea, Romania
2   Department of Environmental Engineering, Faculty of Environmental Protection, University of Oradea, 26 Gen. Magheru Street, 410048 Oradea, Romania
3   Doctoral School of Biomedical Science, University of Oradea, 410087 Oradea, Romania
4   Faculty of Medicine and Pharmacy, University of Oradea, 10 P-ta 1 December Street, 410073 Oradea, Romania
*   Correspondence: vasile.laslo@uoradea.ro (V.L.); svicas@uoradea.ro (S.I.V.)
†   These authors contributed equally to this work.

**Abstract:** This study was conducted to evaluate whether the application of proline as a potential osmoprotectant at different doses could improve the genotoxic and mutagenic effects caused by plant exposure to cadmium salts. For this purpose, the Comet assay was used, which allows the rapid detection of DNA damage shortly after its occurrence, before the DNA is repaired, as well as the discrimination of the DNA damage limited to specific cells in a heterogeneous population. After treatment of *Allium cepa* roots with 75μM $CdSO_4 \cdot H_2O$ (Cd sample), a DNA percentage of 35.24% was recorded in the tail. In the samples treated first with proline and then with cadmium (pre-treatment group), the percentage DNA in the tail was reduced by 24.8% compared with the Cd sample. Instead, in the post-treatment group (samples treated first with cadmium and then with proline), the percentage DNA in the tail was reduced by 69.04% compared with the Cd sample. All cadmium treatments induced chromosomal aberrations (CAs). Compared with the CAs values obtained after Cd treatment, the reduction was 75.6% in the pre-treatment group and 55.39% in the post-treatment group. The results of this study highlighted that exogenous application of proline alleviated the genotoxic effect of cadmium.

**Keywords:** comet assay; chromosomal aberrations; DNA damage; cadmium; proline

## 1. Introduction

The presence of cadmium in ecosystems is mainly due to to human activity but also to the mineralization processes of rocks that contain metals including cadmium [1]. In agricultural soils, the presence of cadmium is mainly due to the administration of phosphorus-based fertilizers [2]. Multiple studies confirm the phytotoxicity of cadmium, which manifests by inducing an oxidative explosion by generating reactive oxygen species (ROS) [3], by perturbing the antioxidant systems [4], by having an inhibitory effect on photosynthesis and cellular division [5], and by inducing necrosis in leaves and roots and chlorosis in leaves [6], while also being capable of perturbing the metabolism of chlorophyll [7], mineral nutrition [8], and water homeostasis [9], affecting transpiration and the fixation of carbon dioxide, and modifying the permeability of cell membranes. Within the soil, cadmium is generally present in mobile form, which has a negative ecological significance. The mobile form determines a relatively large capacity for the migration of the element within the landscape and leads to increased pollution due to the flux of substances from soil to plants [10]. The accumulation of proline inside plants is a general adaptation to stress factors, which has been highlighted under different types of stress (high salinity, low temperatures, lack of nutrients, presence of heavy metals within the environment).

The first reports of proline accumulation in wilted rye tissues were written by Kemble and MacPherson [11]. The normal level of proline in *Arabidopsis thaliana* seedlings is 1 µmol/g fresh mass, and under saline stress conditions (120 mM NaCl), this level increases eightfold. In tobacco leaves (*Nicotiana tabacum*), the proline content increases 20-fold, and in soy (*Glycine max*), it increase 11-fold if plants are exposed to 200 mM NaCl treatment [12]. In plants, proline constitutes less than 5% of the total free amino acids under normal conditions. After the induction of stress, this level can increase to up to 80% of amino acid reserves [13].

The increase in the endogenous proline content as a response to biotic or abiotic stress factors is a mechanism commonly observed in bacteria, algae, and plants [14–17]. The exogeneous application of proline in cell cultures exposed to cadmium salts [18] causes an increase in the activity of antioxidant enzymes, as well as an increase in the tolerance to oxidative stress [19–21]. Stress conditions perturb intracellular redox homeostasis by challenging mechanisms that balance oxidative stress. The role of proline in stress resistance is explained by its properties as an osmolyte and its ability to balance water stress. Plants exposed to various stress conditions have an overproduction of proline, which gives the plant an increased resistance to stress by maintaining osmotic balance for cell turgor and the stability of cell membranes, preventing the leakage of electrolytes and bringing the concentration of ROS to normal values [12]. In addition to acting as an excellent osmolyte, proline plays three major roles during stress, as a metal chelator, an antioxidative defense molecule, and a signaling molecule [22]. In a study performed by Sharma et al. 1998 [23], exogenous proline protected the activity of glucose-6-phosphate dehydrogenase and nitrate reductase in vitro against inhibition by Cd and Zn. This protection was due to the formation of a proline–metal complex. L-Proline is a multifunctional amino acid that plays an essential role in primary metabolism and in the performance of physiological functions [24]. In soy plants under drought and salinity conditions, the accumulation of proline can be 100 times greater than that in plants grown under optimal conditions [25]. The exogenous application of proline increases nitrogen fixation and the content of antioxidant compounds especially phenolic compounds, carotenoids, flavonoids, and tocopherols [26,27]. The accumulation of proline in plants also appears during exposure to UV radiation, heavy metal ions, pathogenic agents, and oxidative stress [21].

Single cell gel electrophoresis (SCGE) or the Comet assay is a sensitive testing system used to evaluate the genotoxic potential of various polluting agents. Comet analysis allows the detection of the DNA damaging capacity of chemical substances in the environment [28–32]. Measuring the comet tails is an important parameter because it represents free DNA fragments and shows lesions within individual cells [33]. Quite often, the capacity of certain polluting agents in the environment (including heavy metals) to damage DNA is proven by the comet assay using species such as *Allium cepa*, *Hordeum vulgare*, *Arabidopsis thaliana*, *Glycine max.*, *Vicia faba,* and *Zea mays* as bio-indicators of genetic toxicity [34,35].

In our study, we used the Comet assay to test the effects of the exogeneous application of proline in diminishing the genotoxic effect of cadmium. The study shows, as a novelty, that treatments with proline can reduce the genotoxic effects of cadmium on plants.

## 2. Materials and Methods

### 2.1. Experimental Design

Treatments were undertaken according to Fedel-Miyasato et al. [36], with some modifications. Bulbs of *Allium cepa* cv. Vidalia with a 3 cm diameter were immersed in sterile water for 72 h, at 24 °C, in darkness, at the level of the radicular meristem. After this time interval, the roots were 2–3 cm long, and two types of treatment were applied, called pretreatment and post-treatment. For both treatments, the following samples were used: the negative control (CTRL−), consisting of the immersion of the onion roots in 1% methanol solution, the positive control (CTRL+), consisting of the immersion of the roots in 15 µM

H$_2$O$_2$ solution, and the Cd sample (Cd), consisting of the immersion of roots in 75 μM CdSO$_4$·H$_2$O solution. In all three samples, the root immersion time was 24 h (Figure 1a,b).

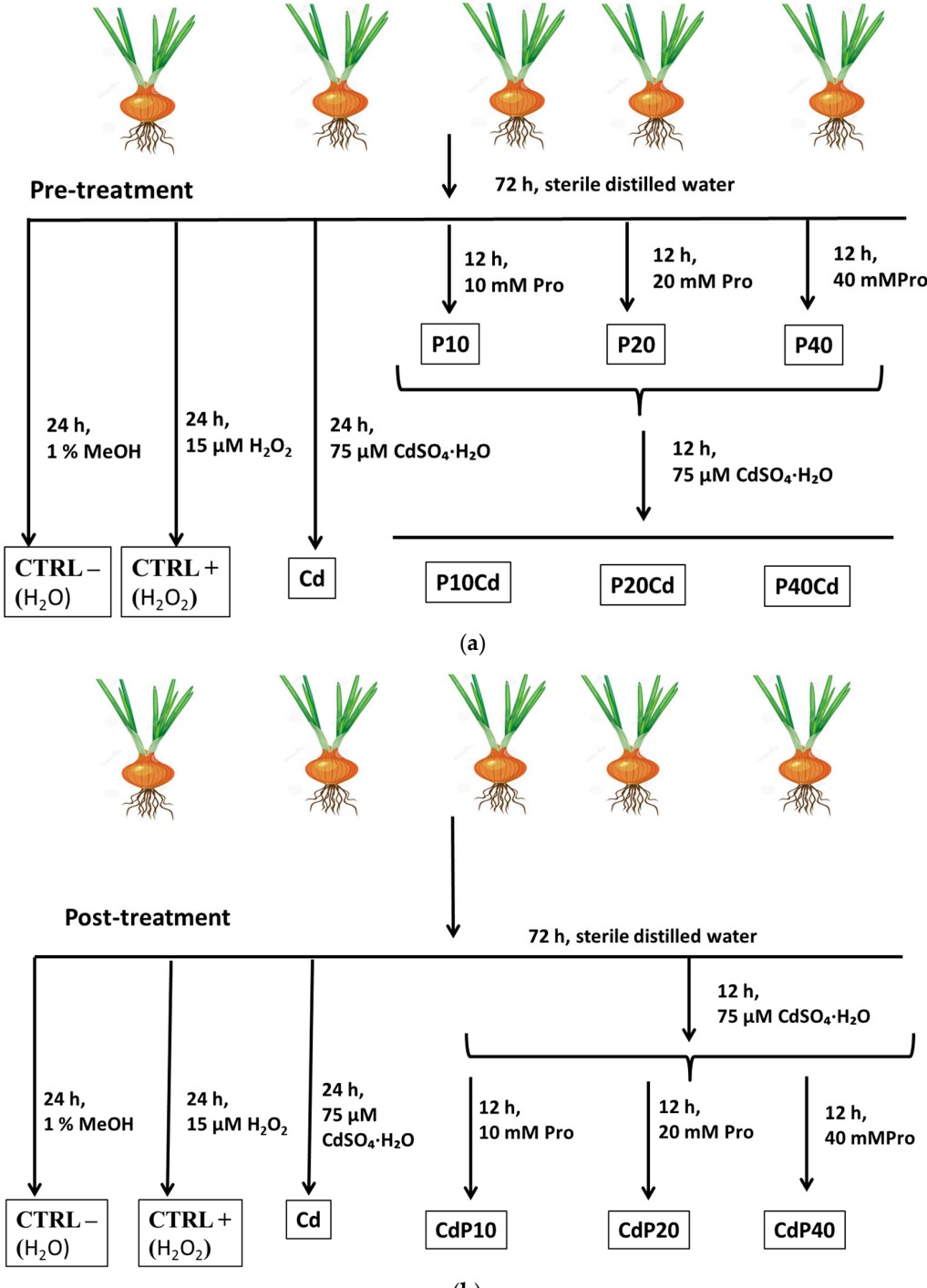

**Figure 1.** Experimental design in the pre-treatment (**a**) and post-treatment (**b**). CTRL—(negative control sample, treatment with 1% methanol); CTRL+ (treatment with 15 μM H$_2$O$_2$); Cd—samples treated with 75 μM CdSO$_4$·H$_2$O; P10, P20, and P40—samples treated with 10, 20, and 40 mM proline, respectively; P10Cd, P20Cd, and P40Cd—samples treated with 75 μM CdSO$_4$·H$_2$O after te treatment with different concentrations of proline (10, 20, and 40 mM, respectively); CdP10, CdP20, and CdP40-samples treated with 75 μM CdSO$_4$·H$_2$O for 12 h and then with 10, 20, and 40 mM proline, respectively.

The pre-treatment consisted of exposing the roots to different proline (Duchefa, CAS No:147-85-3) doses of 10 mM (P10), 20 mM (P20), and 40 mM (P40) for 12 h at room temperature. After this time interval, the roots were washed with distilled water and immersed in the 75 μM CdSO$_4$·H$_2$O solution for another 12 h (Figure 1a). At the end of the pretreatment, the following samples were obtained: P10Cd, P20Cd, and P40Cd, in which the roots were treated with 10, 20, and 40 mM proline, respectively, followed by treatment with cadmium.

In the case of post-treatment, the roots were initially immersed in the 75 μM CdSO$_2$·H$_2$O solution for 12 h, after which they were treated with different concentrations of proline (10, 20, and 40 mM) for another 12 h, obtaining the following samples: CdP10, CdP20, and CdP40 (Figure 1b). At the end of both treatments, the roots were washed, separated from the bulb, and kept on ice while waiting for the core to be extracted

*2.2. Comet Assay*

2.2.1. Preparation of the Slides for the Comet Assay

Standard microscope glass slides were degreased and covered with a 1% normal melting point agarose (NMPA) (Bio-Rad, Hercules, CA, USA) in a phosphate buffered saline, PBS (Lonza–Verviers, Verviers, Belgium).

2.2.2. Preparation of Plant Cell Nuclei

Approximately 25 root tips of 3 mm each were placed in a cold Petri dish, inclined, in 600 μL of ice-temperature Tris-MgCl$_2$ buffer (0,2 M Tris, pH 7.5; 4 mM MgCl$_2$–6H$_2$O; 0.5% g/v Triton X-100). The roots were immediately sliced with a razor, and then the slices were minced for approximately 15 s, according to recommendations by Pourrut et al. [37], in order to release the nuclei and to collect them in the Tris-Mg Cl$_2$ buffer.

2.2.3. Single-Cell Gel Electrophoresis

The protocol for neutral electrophoresis, with some modifications, is that described by Wojewodzka et al. [38]. The neutral Comet assay highlights the double catenary ruptures of the DNA, the comets having well-defined contours, appropriate for image analysis. All operations were done under inactinic red light to avoid DNA photodegradation. Specifically, 80 μL of 0.8% agarose with a low melting point (LMPA) (Thermo Scientific, Waltham, MA, USA) at 37 °C was mixed with 50 μL of nuclear suspension and deposited on a slide pre-covered with 1% normal melting-point agarose (NMPA). The suspension was covered with a 25 × 40 mm glass slide and left to solidify for 5 min on a tray cooled with ice. After solidification, the slides were immersed in ice-cold lysis solution (2.5 M NaCl, 100 mM EDTA, 10 mM Tris-HCl, 1% N-lauroilsarcozine, pH 9.5, to which 0.5% Triton X-100 and 10% dimetilsulfoxide (DMSO) were added before use. After lysis (for 1 h at 4 °C in the dark), the slides were washed three times, 5 min each time, with an electrophoresis buffer, and were placed in the horizontal electrophoresis unit, in the electrophoresis buffer (300 mM sodium acetate, 100 mM Tris-HCl, pH 8.3) for 1 h in the dark at 4 °C. The slides were then subjected to electrophoresis in the dark for an hour at 14 V (0.5 V/cm, 11–12 mA) at 4 °C.

2.2.4. DNA Damage Evaluation

After electrophoresis, the slides were washed three times with cold distilled water, for neutralization, colored with 50 μL ethidium bromide (EtBr) (2 μg/mL), and then covered with a glass slide. From each slide, 50 nuclei were selected randomly, which were analyzed with a Bio Systems fluorescent microscope, equipped with a 546 nm excitation filter and a 590 nm emission filter. Three slides were evaluated per treatment, and each treatment was repeated twice. The images were processed with Comet Score software (Comet Score 2.0.0.38; TriTek Corp., Sumerdock, VA, USA).

### 2.3. Cytogenetic Test

Upon final exposure, five roots from each bulb were immediately fixed for 2 h in Clarke's solution (ethanol, glacial acetic acid; 3:1) and afterwards in 96% ethanol for 15 min. Then, the samples were placed in 70% ethanol at 4 °C. For cytological studies, the roots were hydrolyzed in HCl (1 N) for 15 min at 60 °C, washed with distilled water, and colored for 4 min with 2% aceto-orcein. The cells were displayed under slides (using the squash technique) and were analyzed under a binocular microscope (Typ H 600 LL Helmut Hund GMBH, Wetzlar, Germany) at 400× magnification. For each experimental variant, 5 slides were prepared, and at least 3500 cells were analyzed for each experimental variant. The mitotic index (MI) was determined by using Equation (1):

$$\text{Mitotic index (\%)} = \text{Number of mitotic cells/Total number of cells} \qquad (1)$$

### 2.4. Statistical Analysis

The results represent the mean and standard deviation (SD) of 3 independent experiments. Statistical significance between the samples was determined by one-way ANOVA and Tukey's multiple comparison test, using GraphPad Prism (version 8.01). A *p* value < 0.05 was considered statistically significant.

## 3. Results and Discussion

Results obtained via the Comet assay are reported using different descriptors, of which the most frequently used are the percentage of DNA in the tail, the length of the tail, the moment of the tail (obtained by multiplying the percentage of DNA in the tail by the length of the tail), and the OTM (olive tail moment) [39]. In reporting the results of this study, we chose to use the descriptor of the percentage of DNA in the tail (Figure 2) because this parameter is linearly connected to the rupture frequency [40].

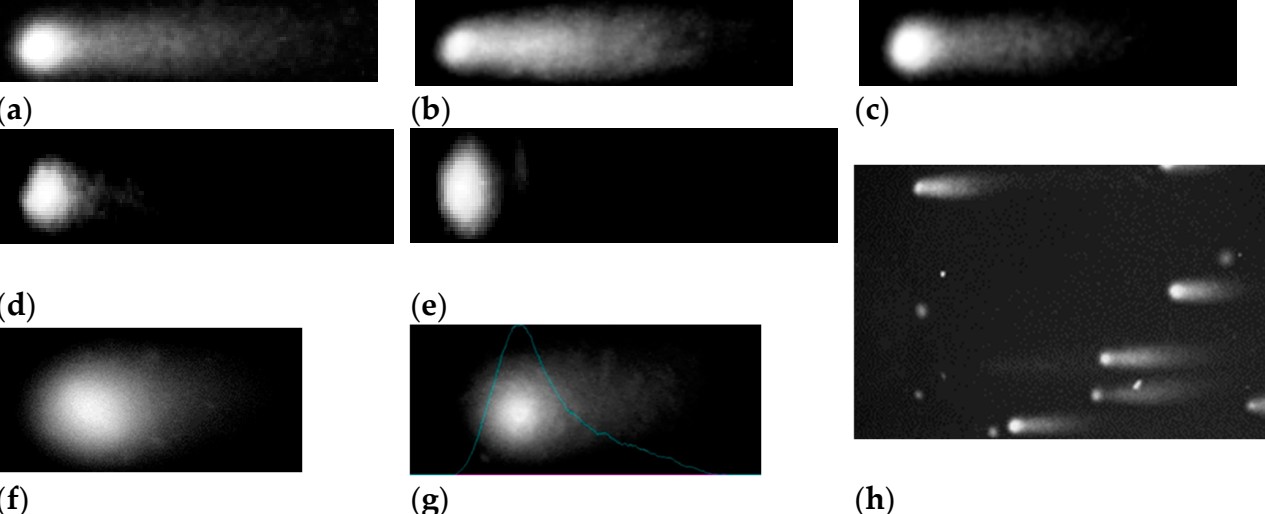

**Figure 2.** Representative images of neutral comets. (**a**) Positive control (CTRL+)—treatment with 15 μM $H_2O_2$ for 24 h; (**b**) Cd—treatment with 75 μM $CdSO_4 \cdot H_2O$ for 24 h; (**c**) Pre-treatment (P40Cd)—samples treated with 40 mM Pro for 12 h and then with 75 μM $CdSO_4 \cdot H_2O$; (**d**) Post-treatment— (CdP40) samples treated with 75 μM $CdSO_4 \cdot H_2O$ for 12 h and then with 40 mM Pro; (**e**) CTRL—(control sample, treated with 1% methanol); (**f**) P10—samples treated with 10 mM proline; (**g**) P40—samples treated with 40 mM proline (**h**) Comet profiles in the neutral Comet test.

Numerous publications that describe the results of comet tests support the assumption that this descriptor offers reliable information on the migration of DNA in the comets [41–44]. Regarding the oxidative-induced DNA deterioration, we found a significant response between the test variants, which confirms observations that cells treated with different concentrations of $H_2O_2$ [45] show broken DNA, a fact that leads to a substantial increase

in the DNA percentage found in the tail [46]. We determined statistically significant DNA deterioration induced by the positive control, with $p < 0.05$ compared to the other variants (Figure 3). An exception is the treatment with 75 μM CdSO$_4$ · H$_2$O, where recorded values, processed via the Tukey test, showed no significant difference (6.05%) with IC 95% from −11.2779 to 0.0258 and with $p = 0.0258$. In a previous screening, we found that the dose of 15 μM H$_2$O$_2$ induced significant DNA damage, which is why we used this dose in the experiment (data not shown).

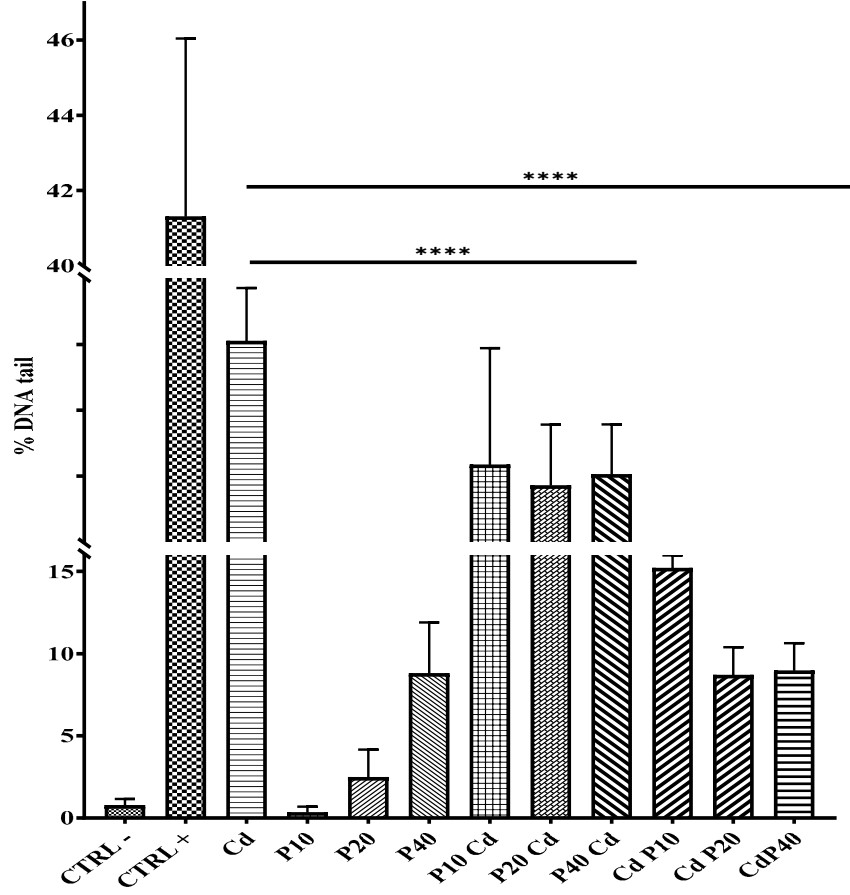

**Figure 3.** Effect of the exposure of *Allium cepa* meristem cells to Cd (Cd sample) and different concentrations of proline (P10, P20, and P40—samples treated with 10, 20, and 40 mM proline, respectively). Pre-treatment (P10Cd, P20Cd, and P40Cd) and post-treatment (CdP10, CdP20, and CdP40). Results are expressed as the mean ± SD and compared the Cd sample with pre-treatment and post-treatment samples. **** $p < 0.00001$ statistically significant differences determined by Tukey's multiple comparison test.

In the case of treatment with CdSO$_4$·H$_2$O (Cd) (Table 1), Tukey's multiple comparison test showed statistically significant differences, with $p < 0.05$, compared to the post-treatment group ($p = 0.0069$ in Cd-CdP10; $p < 0.0001$ in Cd-CdP20 and Cd-CdP40). Cadmium interacts with DNA and manifests its genotoxic effect either directly [47,48] or indirectly [31].The direct genotoxic effect may involve the binding of Cd to DNA, and the indirect interaction may be associated with oxidative damage to DNA, increasing cellular oxidants in the cells [49], and the stimulation of ROS synthesis that damages DNA and causes the manifestation of oxidative stress [50]. The deterioration of DNA in the cells of *Vigna unguiculata* suggests that under Cd toxicity, the production of ROS leads to catenary ruptures and to a series of structural modification to the nucleus [51].

**Table 1.** Statistical significance between all samples (*p* values).

| | CTRL− | CTRL+ | Cd | P10 | P20 | P40 | P10 Cd | P20 Cd | P40 Cd | Cd P10 | Cd P20 | Cd P40 |
|---|---|---|---|---|---|---|---|---|---|---|---|---|
| CTRL− | 1 | <0.0001 | <0.0001 | >0.9999 | 0.9985 | 0.0013 | <0.0001 | <0.0001 | <0.0001 | <0.0001 | 0.0016 | 0.0009 |
| CTRL+ | **** | 1 | 0.0499 | <0.0001 | <0.0001 | <0.0001 | <0.0001 | <0.0001 | <0.0001 | <0.0001 | <0.0001 | <0.0001 |
| Cd | **** | * | 1 | <0.0001 | <0.0001 | <0.0001 | <0.0001 | <0.0001 | <0.0001 | <0.0001 | <0.0001 | <0.0001 |
| P10 | ns | **** | **** | 1 | 0.9895 | 0.0005 | <0.0001 | <0.0001 | <0.0001 | <0.0001 | 0.0006 | 0.0004 |
| P20 | ns | **** | **** | ns | 1 | 0.0318 | <0.0001 | <0.0001 | <0.0001 | <0.0001 | 0.0378 | 0.0242 |
| P40 | ** | **** | **** | *** | * | 1 | <0.0001 | <0.0001 | <0.0001 | 0.0281 | >0.9999 | >0.9999 |
| P10 Cd | **** | **** | **** | **** | **** | **** | 1 | 0.9993 | >0.9999 | <0.0001 | <0.0001 | <0.0001 |
| P20 Cd | **** | **** | **** | **** | **** | **** | ns | 1 | >0.9999 | 0.0001 | <0.0001 | <0.0001 |
| P40 Cd | **** | **** | **** | **** | **** | **** | ns | ns | 1 | <0.0001 | <0.0001 | <0.0001 |
| Cd P10 | **** | **** | **** | **** | **** | * | **** | *** | **** | 1 | 0.0235 | 0.0368 |
| Cd P20 | ** | **** | **** | *** | * | ns | **** | **** | **** | * | 1 | >0.9999 |
| Cd P40 | *** | **** | **** | *** | * | ns | **** | **** | **** | * | ns | 1 |

Statistical significance: * $p < 0.05$; ** $p < 0.01$; *** $p < 0.001$; **** $p < 0.0001$. ns: not significant.

Compared to an average value of 35.24% DNA in the tail recorded during treatment with $CdSO_4 \cdot H_2O$, in the post-treatment group, the % DNA in the tail was reduced by 56.87% in response to CdP10, by 75.51% in response to CdP20, and by 74.71% in response to CdP40. This reduction in the length of the tail is due to the protective effect of applying proline. It is known that the exogenous application of proline increases plant stress resistance by modulating the endogenous metabolism of proline [52]. ROS can lead to impaired physiological function through cellular damage to DNA, proteins, lipids, and other macromolecules [53]. The application of exogenous proline [54] activated the antioxidant systems by increasing the activities of catalase, ascorbate peroxidase, and superoxide dismutase. Quantitative analysis of the present study showed a significant reduction in the percentage of DNA in the tail after pre-treatment with proline, highlighting its de-mutagenic role. Quantifying the percentage of DNA in the tail confirmed its reduction by 26.7% in the P10Cd variant, by 31.15% in the P20 Cd variant, and by 28.77% in the P40Cd variant. In both tested situations (pre- and post-treatment with proline), we recorded a significant reduction in the %DNA in the tail, compared to the $H_2O_2$ treatment and $CdSO_4 \cdot H_2O$ treatment. Comparative analysis of the % DNA in the tail between the P40 sample with the Cd P20 and Cd P40 post-treatment samples revealed no statistically significant differences (Table 1), the DNA repair process being able to occur even in the absence of antigenotoxic substances. However, the experimental design we used together with the specific requirements of the method (working under yellow light, low temperature and execution speed) ensured that real results were obtained. The accumulation of excessive concentrations of heavy metals in plants generates stress that leads to serious physiological and structural disturbances [22]. One of the responses of plants to this state of stress is the accumulation of a large amount of proline in the cells [55,56]. Proline acts as both an osmoprotectant and a heavy metal chelator by inducing the formation of phytochelatins that chelate with heavy metals such as Cd, thereby reducing their toxicity [57]. Our observations confirm this role of proline, as in all of the experimental variants where cadmium was associated with proline, its level of genotoxicity was reduced compared to the variants treated with cadmium alone. It was reported [58] (Xu et al., 2009) that the exogenous application of proline decreased ROS levels, improving cadmium tolerance. According to Roy et al. 1993 and Jain MJ et al., 2003 [59,60], the effect of exogenously applied proline is dependent on its concentration. In our study, the treatments with 10 and 20 mM proline solutions generated lower percentages of DNA in the tail than the treatment with 40 mM proline solution.

According to Kada et al. [61], there are two classes of protective substances that act against DNA damage: antimutagenic compounds that block the action of agents that

induce DNA deterioration, mainly by absorption, and that act preferentially on extracellular mediums; and bio-antimutagens capable of preventing DNA lesion formation or being involved in the modulation of DNA repair. According to our results, in the pre-treatment group (where the roots were exposed to proline for 12 h and then to Cd, a reduction in DNA de-structuring was observed, and therefore an antimutagenic quality can be attributed to proline [62], although it exerts this function indirectly by modifying the activity of antioxidant enzymes [63]. Statistical analysis of the values of the negative control (CTRL-) compared to the values of the P10, P20, and P40 samples revealed no statistical differences, except for P10 vs. P40, where a significant difference was recorded ($p = 0.43$).

From the data presented in Table 2, it was observed that cadmium had a genotoxic effect in all samples in which *Allium cepa* roots were treated with this metal. The results we recorded are in agreement with the results obtained by other authors [32,64], who attribute this mitotic depression and %MI reduction to the fact that the heavy metal prevents cells from entering prophase, thus blocking the mitotic phase of the cellular cycle. Frequently, this induces modifications in chromosome structures, which manifest as chromosome ruptures (Figure S1a,f), delayed chromosome formation (Figure S1b), unequal distribution of chromatin (Figure S1c), and the formation of bridges (Figure S1d,e). In all experimental variants in which the Cd treatment was accompanied by pre- or post-treatment with proline, the genotoxic effect of cadmium was significantly diminished ($p \leq 0,05$) compared to the positive control and to the variant to which the only treatment administered was cadmium (Table 2).

**Table 2.** Mitotic index (MI) and chromosomal aberrations (CAs) % in meristem cells of *Allium cepa* roots exposed to Cd and proline depending on treatments.

| Treatment | MI (%) | Interphase | Prophase | Metaphase | Anaphase | Telophase | CAs (%) |
|---|---|---|---|---|---|---|---|
| CTRL− | 64 ± 1.8 [a] | 1260 | 1826 | 224 | 114 | 84 | 1.6 ± 0.14 [cd] |
| CTRL+ | 2.4 ± 0.22 [e] | 3416 | 42 | 23 | 12 | 7 | 4.2 ± 0.30 [b] |
| Cd | 2.8 ± 0.37 [e] | 3402 | 48 | 19 | 15 | 16 | 5.2 ± 0.58 [a] |
| P10 | 40 ± 1.75 [b] | 2100 | 386 | 256 | 155 | 98 | 0.0 |
| P 20 | 40 ± 1.49 [b] | 2083 | 870 | 278 | 186 | 65 | 0.0 |
| P 40 | 38.1 ± 1.21 [b] | 2170 | 882 | 248 | 173 | 32 | 0.0 |
| **Pre-treatment** | | | | | | | |
| P10Cd | 13 ± 0.82 [c] | 3045 | 196 | 113 | 101 | 45 | 1.2 ± 0.18 [c] |
| P20Cd | 15.4 ± 0.58 [c] | 2975 | 214 | 146 | 114 | 51 | 1.4 ± 0.16 [c] |
| P40Cd | 16 ± 1.12 [c] | 2938 | 210 | 168 | 132 | 50 | 1.2 ± 0.24 [c] |
| **Post-treatment** | | | | | | | |
| CdP10 | 8.4 ± 0.74 [d] | 3200 | 146 | 79 | 48 | 21 | 2.1 ± 0.23 [d] |
| CdP20 | 9.4 ± 0.43 [d] | 3169 | 172 | 98 | 40 | 19 | 2.4 ± 0.34 [d] |
| CdP40 | 9.6 ± 0.59 [d] | 3164 | 194 | 106 | 32 | 4 | 2.4 ± 0.44 [d] |

Different letters for each sample denote statistically significant differences ($p = 0.05$) between pre- and post-treatment samples and Cd samples, determined by Tukey's multiple comparison test. Results are expressed as the mean ± sd.

Statistically significant differences in %MI were also recorded in pre- and post-treatment groups according to the concentration of proline administered (10, 20, 40 mM). Recent research by Hassan et al. [65] highlighted the role of proline as an attenuating agent that reduces mutagenic effects of Cd by increasing %MI and decreasing CAs in comparison with untreated samples (CTRL-, P10, P20, and P40). All treatments with cadmium induced CAs. Compared with CAs values obtained after Cd treatment, the reduction was 75.6% in the pre-treatment group and 55.39% in the post-treatment group.

## 4. Conclusions

The Comet test may be used in in situ monitoring of plants exposed to heavy metals. Through measurements of the DNA lesions, the genotoxic effects of cadmium under low levels of exposure can easily be evaluated. The exogenous application of proline alleviated the genotoxic effect of cadmium. This fact suggests that proline treatments could be applied to plants grown on cadmium-containing soils. Plants grown on cadmium-polluted soils can be treated with proline to reduce the effects of stress and to stimulate the production of biomass for energy purposes. Future studies are needed to identify the mechanisms by which proline annihilates the toxic effect of heavy metals.

**Supplementary Materials:** The following supporting information can be downloaded at https://www.mdpi.com/article/10.3390/agriculture12101568/s1, Figure S1: Chromosomal aberrations in meristem cells of *Allium cepa* exposed to 75 µM CdSO4·H2O. (a) lagging chromosome formation; (b) aberrant nucleus; (c) unequal distribution of chromatin; (d,e) bridge formation; (f) chromosome displacement at anaphase.

**Author Contributions:** Conceptualization, V.L. and C.P.; methodology, V.L. and S.I.V.; validation, C.P. and A.R.M.; formal analysis, V.L. and S.I.V.; investigation, V.L., C.P., E.A., F.M. and A.R.M.; resources, V.L. and C.P.; writing—V.L. and C.P.; writing—review and editing, V.L. and S.I.V.; visualization, A.R.M., F.M. and E.A.; supervision, V.L.; project administration, V.L. and S.I.V.; funding acquisition, E.A. All authors have read and agreed to the published version of the manuscript.

**Funding:** This research received no external funding.

**Institutional Review Board Statement:** Not applicable.

**Data Availability Statement:** Not applicable.

**Acknowledgments:** The authors thank to University of Oradea, Romania for the financial support in publishing this paper.

**Conflicts of Interest:** The authors declare no conflict of interest.

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
