# Peer review of "Antigenotoxic and Antimutagenic Potentials of Proline in Allium cepa Exposed to the Toxicity of Cadmium"

_agriculture, doi:10.3390/agriculture12101568_

Round 1

Reviewer 1 Report

The authors of the manuscript raise an interesting research topic, trying to answer important questions and designing the experiment correctly. However, the manuscript requires text editing and some information supplementation before publication. The comments below are recommended to be taken into account when revising the manuscript.

Line 20 and 21: it is “in the pre-treatment group” and „the 20 post-treatment group” –  it should briefly stated how the groups were acquired and how they were treated

Abstract: a concluding sentence is missing

Introduction: More emphasis should be placed on the protective role of proline under stressful conditions. Proline not only acts as an osmolyte, its role is much broader and it should be supplemented and expanded.

Line 21: it is “is due mainly” it  should be rather “is mainly due to”

Line 33: it is ”an inhibitive effect” it should be rather “an inhibitory effect”

Line 34: it is “an chlorosis” it should be rather “and chlorosis”

Line 36:  it is “affecting perspiration” it should be rather “affecting transpiration”

Line 46: it is “NaCl: 120 mM” it should be rather “120 mM NaCl”

Line 45, 47, 100, 116, 119, 158 and others: a lot of spaces are missing e.g. “is1 μmol/g”, “Nicotianatabacum”, “Tris–MgCl2buffer”, “Wojewódzkaet. al [36]”, “μLofagarose”, “OTM(olive”

Line 54: it is “the increase of antioxidant enzyme activity” it should be rather “the increase in the activity of antioxidant enzymes”

Line 57: it is “hydric stress” it should be rather “water stress”

Line 59-60: it is “during drought and salinization conditions” it should be rather “ under drought and salinity conditions”

Line 62: it is “antioxidant compound content, specifically” it should be rather “the content of antioxidant compounds, in particular”

Line 75: it is “ the Allium cepa assay” it should be rather “ the Comet assay”

Materials and Methods: a name of cultivar of Allium cepa is missing

Line 84 nad 85: it is ”the negative control group …. in the positive control“ –  it is not clear why and for what purpose these two groups were obtained

Line 86: it is “H2O2 -15 µM” should be rather “ 15 µM H2O2”.

Line 94: “The steps followed in the two treatmentsand the coding of the samples obtained” - the sentence is incomprehensible

Figure 1: the diagram does not explain well the design of the experiment, the legend of the figure describes the design of the experiment better than the figure itself

Line 106: it is “Regular slides”, it should be rather “standard microscope glass slides ”

Line 106: “(NMPA)1%”-  ?

Line 118: „deterioration” - of what?

Line 127: “within the electrophoresis buffer” -> “in the electrophoresis buffer”

Line 143: its “2% acetic orcein”, recommended is rather “2% aceto-orcein”.

Line 155: it is “via comet assay”, should be “via the comet assay”

Line 156” it is “the percentage of tail DNA”, should be rather “the percentage of DNA in the tail”

Line 174: it is” the oxidative induced DNA deterioration” its should be “the oxidative-induced DNA deterioration”

Line 194-195: it is “or via mediation” - ??

Line 210: it is “the activity of the P5CS gene” - as you know genes do not show activity, activity is a feature of enzyme proteins encoded by genes

Line 210: it is “which codifies”, should be ”which encodes”

Line 271: “the Allium test” rather should be “the Comet test”

Reviewer 2 Report

The study concerns and interesting topic of metals genotoxicity and includes the novelty concerning the protective effects of proline. The methodology is well described and the inclusion of both pre- and post-treatment with proline adds value to the obtained results.  

However, there are some concerns, which impede the publication of the results:

-          I am not sure if the study is directed to the right journal as it has very vague connection with agriculture

-          The Table 2 lacks control, it includes only a positive control

-          The interpretation of the results could be improved. E.g. is it possible that the observed reduction in %DNA in tail in post-treatment is associated with repair mechanism independent from the action of proline?

Is it possible that pre-treatment with proline affects Cd uptake and thus also attenuates its genotoxic effects. In my opinion Cd level should be measured in Cd-treated and Cd and proline pre- and post-treated roots.

How can you explain the increase in %DNA in tail in response to proline only (Fig. 3)? In P40 it actually reaches the level of CdP40 so it is difficult to distinguish between proline and Cd effects.

Other comments:

Lines 19-20: it should be specified that pre- and post-treatment refers to proline.

In Fig. 1 it would be good to place the b section below the a to increase the resolution and clarity of the figure.

Figure 2: I would advice to include the representative images of all of the experimental variants.

Lines 194-195: It is an oversimplification. There are also other reasons for Cd-inducible over-accumulation of ROS.

Lines 202-207: As this is a post-treatment is it possible that the reduction in %DNA in tail is related to the activation of repair systems independently from the proline action?

Lines 209-213: What is the connection of the cited study with that part of the discussion?

Line 213: It would be good to highlight that the sentence “quantitative analysis showed” refers already to the results of present study

Line 249: It should be compared with Cd treatment rather that with the positive control.

Fig. 4: The quality could be improved.

In conclusion: The second sentence is difficult to follow.

In conclusions: Would it be economically feasible to apply proline in the fields? The plants could grow better due to attenuation of genotoxic effects but they would probably still contain metals affecting human health.

Reviewer 3 Report

There are numerous typo issues such as spacing and spelling issues in addition to grammar issues.

Audience may like to see the Cd toxicity symptoms of adult plants at the concentrations used in this experiment, although the manuscript focused on the antigenotoxic effects of proline on Cd toxicity, and roots were used mainly for analyses.

Also, it will enhance the quality of the manuscript if it includes at least one antioxidant enzyme analysis.

Round 2

Reviewer 2 Report

Thank you for all the clarifications. Just two minor points:

-          If the resolution of Fig. 4 can not be improved I would suggest to transfer it to supplementary materials. It is though just a suggestion.

-          I would suggest to delete the sentence “By applying exogenous proline, plant-water relations are modulated by maintaining, turgor of cells exposed to stress and by increasing the rate of photosynthesis.” From the conclusions as it is not really related to the obtained results.

Author Response

The authors thank you for your time to review our manuscript.

Reviewer 3 Report

The manuscript reads well and the revision was adequately made.
